# Shoulder Injury Related to COVID-19 Vaccine Administration: A Case Series

**DOI:** 10.3390/vaccines10040588

**Published:** 2022-04-12

**Authors:** Korakot Maliwankul, Pattira Boonsri, Prapakorn Klabklay, Chaiwat Chuaychoosakoon

**Affiliations:** 1Department of Orthopedics, Faculty of Medicine, Prince of Songkla University, 15 Karnjanavanich Road, Hat Yai, Songkhla 90110, Thailand; koopy_medy@hotmail.com (K.M.); pglabgly@yahoo.com (P.K.); 2Department of Radiology, Faculty of Medicine, Prince of Songkla University, 15 Karnjanavanich Road, Hat Yai, Songkhla 90110, Thailand; bpattira@medicine.psu.ac.th

**Keywords:** complication, COVID-19, shoulder injury, vaccination

## Abstract

Background: A shoulder injury related to vaccine administration (SIRVA) is a vaccination complication that can affect daily life activities. To date, there have been no case series of patients diagnosed as SIRVA following a COVID-19 vaccination. We offer a series of seven SIRVA cases including clinical presentations, investigations and treatment outcomes. Methods: A retrospective chart review was performed for seven patients who developed SIRVA following a COVID-19 vaccination between April 2021 and October 2021. All patients had no prior shoulder pain before their vaccination and then developed shoulder pain within a few days following the vaccination, which did not spontaneously improve within 1 week. Results: Four of the seven patients were male, and the average age was 62.29 ± 7.76 years. The average body mass index was 25.1 ± 2.2 kg/m^2^. In all cases, the cause of the SIRVA was from an incorrect COVID-19 vaccine administration technique. Two patients developed shoulder pain immediately following the injection, one patient about 3 h after the injection, and the other four patients within the next few days. Two of the seven patients visited the orthopedic clinic after the persistent shoulder pain for 3 and 4 days and the other five patients 1–9 weeks following their injections. One of the seven patients was treated with combined intravenous antibiotic and oral non-steroidal anti-inflammatory drug (NSAID) because septic arthritis of the shoulder could not initially be ruled out, and recovered within 2 weeks. The other six patients had shoulder pain without acute fever, and five of them were treated with only oral prednisolone 30 mg/day for 5–10 days, following which the pain improved and they all could return to normal activities within 14 days, with no side effects from the prednisolone such as stomachache, nausea, vomiting, headache, or dizziness. Discussion and conclusion: In our series, the most common cause of SIRVA was an incorrect vaccination technique. Most patients responded well to oral NSAIDs or oral prednisolone. Clinical relevance: All SIRVAs were from an incorrect injection technique and not actually the vaccination, so our series highlights the importance of ensuring all vaccinators understand the importance of taking proper care with the injection technique. Additionally, most of our patients with SIRVA from a COVID-19 injection responded well to oral prednisolone (30 mg/day). If there are no contraindications, we suggest this as the first line treatment for COVID-19-related SIRVA.

## 1. Introduction

The COVID-19 pandemic has had a great effect on normal life activities worldwide. To aid the return to normal life activities, one of the important key factors has been the COVID-19 vaccines, which have been widely applied for immunization since early 2021. However, there have been some adverse effects after the COVID-19 vaccinations, which can be divided into local and systemic adverse effects. Most of these effects usually spontaneously resolve within a few days after the vaccination. If the clinical symptoms persist for more than two days, the doctor should be concerned about more serious complications. A shoulder injury related to vaccine administration (SIRVA) is one complication that begins within 48 h following a vaccination, with the main symptoms of shoulder pain with limited range of motion. The pathogenesis of SIRVA is still uncertain, but a widely accepted theory is that injecting the vaccine into the subdeltoid bursa produces a prolonged inflammatory response [1,2,3]. This complication affects daily life activities such as eating, bathing, dressing, etc. The most common cause of this complication is an incorrect vaccine administration technique, such as an incorrect needle direction or using an incorrect landmark for the injection.

To date, there have been no case series published in the literature on patients diagnosed as SIRVA following a COVID-19 vaccination. In this case series, we present seven cases of SIRVA including clinical presentations, physical examinations, investigations, medications, and treatment outcomes after treatment with some combination of oral prednisolone, oral non-steroidal anti-inflammatory drugs (NSAIDs), intraarticular steroid injection, and/or intravenous antibiotic.

## 2. Materials and Method

In our institution we had seven patients between April 2021 and October 2021 who were diagnosed as SIRVA according to the definition of the National Vaccine Injury Compensation Program (VICP) by one of two sports medicine orthopedists. The VICP definition is “shoulder pain with limited range of motion within 48 h after vaccination with no prior history of pain, inflammation, or dysfunction of the affected shoulder before vaccine administration” [1]. This case series was based on data collected by chart review.

## 3. Case Series

During the study period, seven patients visiting the orthopedic clinic of our hospital with clinical symptoms of shoulder pain were diagnosed as SIRVA, as they had had no prior shoulder pain before their vaccination and the shoulder pain developed quickly or within a few days after the vaccination and did not spontaneously improve within 1 week. The patients’ characteristics are shown in Table 1. Four of the seven patients were male, and the average age was 62.29 ± 7.76 years. The average body mass index (BMI) was 25.1 ± 2.2 kg/m^2^. In all cases, our investigations found that the cause of the SIRVA was from an incorrect COVID-19 vaccine administration technique. The cause of the SIRVA in three of the seven patients was use of an incorrect landmark (thus resulting in the injection being given too high on the arm) for the injection, and the cause in the other four cases was from an incorrect direction of the needle causing the vaccine to be injected into the bursa rather than the deltoid muscle. Two patients developed their shoulder pain immediately following the injection, one patient about 3 h after the vaccination, and the other four patients within the next few days. Two of the seven patients visited the orthopedic clinic after persistent shoulder pain for 3 and 4 days and the other five patients 1–9 weeks after their injections. One of the seven patients was treated with combined intravenous antibiotic and oral NSAID because septic arthritis of the shoulder could not initially be ruled out and recovered within 2 weeks. The other six patients had shoulder pain without acute fever, and five of them were treated with only oral prednisolone 30 mg/day for 5–10 days following which the pain improved and they all could return to normal activities within 14 days, with no side effects from the prednisolone such as stomachache, nausea, vomiting, headache, or dizziness. The 7th patient received combined oral prednisolone and an intraarticular steroid injection and recovered in 7 days.

The first case was a 52-year-old Thai male without prior shoulder pain who developed right shoulder pain on day 2 after his Sinovac vaccination. In this case, the injecting nurse used a 1.5-inch 27-gauge needle, with an injection site 3 fingerbreadths below the mid-lateral edge of the acromial process. The direction of the needle was oblique to the skin cephalad (Figure 1 and Figure 2A). He began to experience severe and persistent shoulder pain on day 2 which did not improve, and he came to the orthopedic clinic 6 days after his injection. At that time, the physical examination found pain on all motion directions of his right shoulder and a low-grade fever. Initially, with these symptoms, we could not rule out septic arthritis of the right shoulder, so he was sent for Magnetic Resonance Imaging (MRI) which showed subacromial bursitis (Figure 3A,B). Some yellowish fluid was aspirated from the subacromial bursa, and the fluid analysis showed a white blood cell count of 45,500 cells/mm^3^, monocytes 1%, polymorphonuclear neutrophils 99%, red blood cell count 23,400 cells/mm^3^, and no crystals or organisms. The fluid was sent for culture which showed no organisms. These findings were indeterminate, so he was admitted and treated with combined intravenous cefazolin 1 gm every 6 h for 3 days and oral celecoxib (400 md/day) for 7 days, when the intravenous cefazolin was changed to oral cephalexin (1000 mg/day) for a further 7 days. One day after admission for the intravenous antibiotic, his fever resolved, and he was discharged on day 4 after switching to an oral antibiotic. Over the next few days his symptoms gradually improved, and he could move his shoulder with a full range of motion in all directions 2 weeks after the initial diagnosis and treatment.

The second case was a 51-year-old Thai female with no history of shoulder pain. She received an Oxford-AstraZeneca COVID-19 vaccine with a 1.5-inch 25-gauge needle, based on the landmark of 1 fingerbreadth below the midlateral edge of the acromial process (Figure 2B). The direction of the needle was perpendicular to the skin. Three hours later she began to feel right shoulder pain with limited range of motion. Four days later the pain had not improved and she came to the orthopedic clinic. A physical examination showed pain with limited range of motion in all directions. She was sent for MRI which showed combined subacromial-subcoracoid bursitis (Figure 4A,B). She was treated with oral prednisolone (30 mg/day) for 5 days, and the pain and range of motion improved within 3 days.

The third case was a 66-year-old Thai male who began to have right shoulder pain immediately after a second dose of AstraZeneca. The injection was given with a 1.5-inch 25-gauge needle. The injection landmark was 3 fingerbreadths below the mid-lateral edge of the acromial process. The direction of the needle was 30◦ cephalad to the skin (Figure 2A). The pain was severe enough to disturb his normal life activities, and when it did not decrease after 7 days he came to the orthopedic clinic. The physical examination showed tenderness over the deltoid area and limited range of shoulder motion in all directions. He was diagnosed as SIRVA and initially treated with oral prednisolone (30 mg/day) for 5 days. His clinical symptoms began to improve 6 h after beginning the medication, and he had limited movement in his shoulder in 2 days and full return to normal functions in 7 days.

The fourth case was a 71-year-old Austrian male who began to feel left shoulder pain one day after a second dose of AstraZeneca vaccine. The injection was given with a 1.5-inch 25-gauge needle and the landmark was 1 fingerbreadth below the mid-lateral edge of the acromial process (Figure 2B). The needle direction was perpendicular to the skin. Right shoulder pain was the only symptom, as he had full range of shoulder motion. His pain did not improve after 6 weeks and he finally came to the orthopedic clinic. An MRI of his right shoulder showed a thin layer of subacromial-subcoracoid bursitis and a low-grade partial tear of the supraspinatous tendon (Figure 5A,B). He was given oral prednisolone (30 mg/day) for 10 days and his pain began to improve after 1 day with full return to normal activities in 2 weeks.

The fifth case was a 68-year-old Thai female who began to feel right shoulder pain with limited range of motion 24 h after a second dose of AstraZeneca vaccine. She had had no shoulder pain before the vaccination. The vaccination landmark was 3 fingerbreadths below the acromial process and the needle direction was 45 degrees cephalad to the skin. Her symptoms had persisted for 14 days without improving. They were worse at night, and she could not lay on her right shoulder. She decided to see an orthopedist and the clinical examination showed tenderness at the deltoid muscle. She had a full range of shoulder motion, but with pain at some points. She was sent for ultrasonography which showed a thin layer of subdeltoid bursal fluid and a partial thickness tear of the subscapularis tendon (Figure 6A,B). She was treated with oral prednisolone (30 mg/day) for 5 days and her symptoms improved a week later.

The sixth case was a 64-year-old Thai male who had received his first dose of AstraZeneca vaccine. The vaccination landmark was 3 fingerbreadths below the acromial process with a needle direction of 30 degrees cephalad to the skin. He had developed right shoulder pain with limited range of motion in all directions within 48 h of receiving the vaccine. He did not take any medications, and when the symptoms had not improved in a month he decided to see an orthopedist. A physical examination showed tenderness at the deltoid muscle with limited range of motion in all directions. An ultrasonography showed tenosynovitis at the long head of the biceps and a low-grade partial tear of the subscapularis tendon (Figure 7A,B). He was treated with oral prednisolone (30 mg/day) for 5 days and an intraarticular steroid injection (triamcinolone acetate (TA) 40 mg/mL). His shoulder pain gradually improved, and he could return to normal activities within 7 days after treatment.

The seventh case was a 64-year-old Thai female who presented with right shoulder pain 9 weeks after a first dose of the AstraZeneca vaccine. The symptoms had begun immediately after the vaccination and gradually gotten worse, to the point that it was painful to complete daily activities and she could not sleep on her right side. In her case, the injection landmark was 1 fingerbreadth below the mid-lateral edge of the acromial process. At the OPD, the physical examination showed limited range of right shoulder motion and pain on the forward motion elevation resistance test and external rotation. Ultrasonography showed calcific tendinopathy of the supraspinatous tendon without bursitis (Figure 8A,B). She was given oral prednisolone (30 mg/day) for 5 days, and she had completely recovered at 1 month.

## 4. Discussion

We report a series of seven patients who developed a SIRVA after a COVID-19 vaccination, all of which were the result of an incorrect COVID-19 vaccination technique. Following their examinations, four of the five patients who received only oral prednisolone had rapid improvement, and one patient who received combined oral prednisolone and TA injection also had rapid improvement. One patient received an antibiotic and NSAID due to fever with the shoulder pain, following which he gradually improved over the next 14 days and the 7th patient, who received only oral prednisolone also improved over 14 days. In cases of SIRVA, if the patient has no contraindications such as fever, we recommend giving oral prednisolone (30 mg/day) for at least 5 days.

SIRVA is a rare complication that can occur following a vaccination. The risk factors for this complication are BMI, sex, age and type of vaccine. One study reported that overweight patients had an increased risk of SIRVA [2], while other studies have reported that SIRVA was associated more with incorrect injection techniques [3,4,5] than BMI, similar to our study, in which the average BMI was 25.1 ± 2.2 kg/m^2^. The other factor which has been reported as associated with SIRVA is sex. Two large series reported that SIRVA was predominant in females with incidences of 82.6% and 82.8% [3,6]. In our study, however, only 3 of the 7 patients were female. Older age has also been identified as a risk factor for SIRVA. Studies by Hesse et al. [3] and Atanasoff et al. [2] reported average SIRVA ages of 51 and 50 years, respectively, with only one patient aged below 18 years in each study. The average age in our study was 62.29 ± 7.76 years. This factor could be related to preexisting shoulder pathology and/or insufficient muscle thickness. In children, vaccinations are often completed subcutaneously, and thus there is no chance of overpenetration [2]. Not only specific patient factors can affect the risk of a SIRVA but also the type of vaccine. Two studies have reported that the influenza vaccines were predominant in SIRVA cases at rates of 62% and 84% [2,6]. The influenza vaccine is widely used with more than 150 million doses per year. During the COVID-19 pandemic, however, the COVID-19 vaccines have become the most commonly administered vaccines in the world, thus, it is not surprising that the incidence of SIRVA associated with these vaccines has also become significant.

However, the most common cause of SIRVA, as with most other vaccines, remains an incorrect injection technique. The recommended vaccine injection technique for the COVID-19 vaccines involves three things. First, the landmark should be 2 or 3 fingerbreadths, depending on the size of the patient, below the acromial process (Figure 2C). If the injection landmark is higher than this, there will be a chance of injection into the subdeltoid-subcoracoid bursa or adjacent nonmuscular tissue [4], while if the injection site is lower than this point, there will be a risk of iatrogenic axillary nerve injury. The distance of the axillary nerve from the mid-lateral edge of the acromial process is 52.20 ± 4.21 mm in the arm-at-side position and 49.66 ± 4.54 mm in the 30 degrees of arm abduction position [7]. The second important component of the injection technique involves the direction of the needle, which in all cases should be perpendicular to the skin (Figure 2C). If the injection is given with the needle in the cephalad direction, there will be a chance of penetrating the subdeltoid bursa, leading to a risk of causing iatrogenic subdeltoid bursitis [5].

The main symptom of SIRVA is shoulder pain beginning a few minutes to a few days after the vaccination which does not resolve by a week or so. This is different from the quite common local effect of a vaccination which is usually not severe and clears up in 2–3 days. The key factor in considering SIRVA is the absence of prior shoulder pain before the vaccination. All patients in our study met these conditions, and all complained of difficulty in performing their normal daily activities. There have been reports of neurological symptoms such as weakness or numbness, but these were not found in our study [2,3].

The most common physical examination finding of SIRVA is pain while moving the shoulder with incidences of 55.9% and 85% in reported studies [2,3]. The direction of motion instigating such pain is not specific, with Atanasoff et al. [2] reporting shoulder pain in flexion and abduction while Cook [8] reported greater pain in internal and external rotations. In our study, all patients had shoulder pain in all directions of motion. One male and one female patient had full ranges of motion, but both felt pain during the motions.

Not all patients with a SIRVA are sent for further investigations such as radiography, ultrasonography and/or MRI. Atanasoff et al. [2] reported that 54% of their patients were sent for radiographs and 69% for MRI, while the study of Hesse et al. [3] found that 55.7% and 80.7% were investigated with radiography and MRI, respectively. The MRI findings in these studies were tendinitis or tendinosis (49%), complete or partial rotator cuff tear (44%), fluid collection in muscle (39%), bursitis (34%), joint effusion (10%), muscle edema (39%), bone edema (9.6%), and 5% normal. In our study, six patients were sent for ultrasonography and/or MRI, with findings of bursitis (4 of 6 patients), partial thickness tear of the rotator cuff tendon (4 of 6), and calcific tendinopathy (1 of 6). In our investigations, we found shoulder pathology in five of our cases, which may or may not have been there before their vaccinations. All developed shoulder symptoms within 2 days after their vaccine injections, and therefore it is possible that these patients may have suffered vaccine-induced synovial inflammation aggravating a preexisting pathology [9].

There have been several modalities for treating SIRVA reported in case series, including nonsteroidal anti-inflammatory drugs, lidocaine patches, opioid medications, intra-articular steroid injections, oral steroids, acupuncture and even surgery [2,3,6]. A SIRVA should normally be first treated conservatively, which has resulted in symptom improvement in less than 1 to more than 6 months [1,3]. Some patients have eventually required surgery to relieve their SIRVA due to failure of conservative treatment [3,6]. In our case series, conservative treatment with oral prednisolone, oral NSAIDs and/or intravenous antibiotics was successful in all patients. Prednisolone is a corticosteroid which reduces inflammation and can be used to treat many diseases including allergies, skin diseases, infections, and autoimmune diseases, is also used to help prevent organ rejection following transplantations and has been shown to be effective for SIRVA. A short course of oral prednisolone up to 6 weeks can also be helpful in many situations without harmful effects [10]. Six of the seven cases in our study were given oral prednisolone (30 mg/day) for 5–10 days and none of our patients had side effects such as stomachache, nausea, vomiting, headache or dizziness. If the orthopedist is concerned about using oral prednisolone, NSAIDs are also effective in treating this condition.

## 5. Conclusions

All SIRVAs following a COVID-19 vaccination in this series were the result of an incorrect COVID-19 vaccination technique. Most patients responded well to a short course of oral prednisolone.

## Figures and Tables

**Figure 1 vaccines-10-00588-f001:**
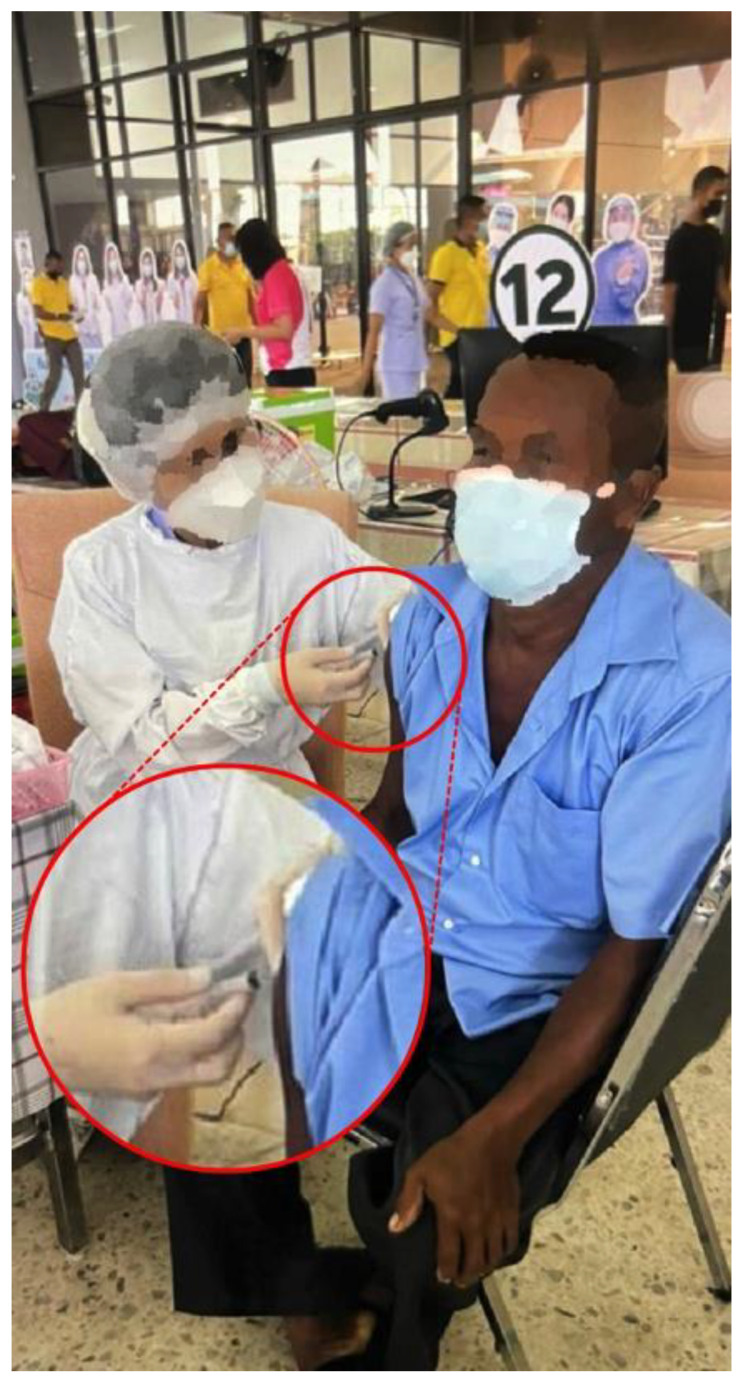
Photograph of the injection being given to Patient 1.

**Figure 2 vaccines-10-00588-f002:**
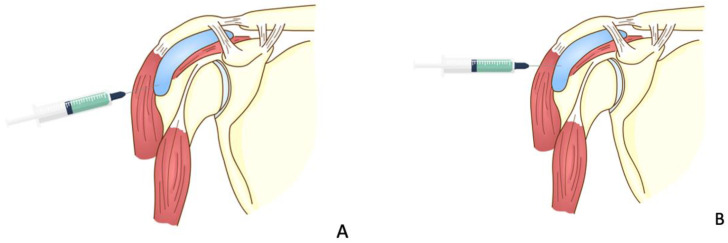
The injection techniques. (**A**) The needle direction is incorrectly cephalad to the skin, (**B**) the incorrect landmark is being used for the injection, (**C**) correct direction and landmark.

**Figure 3 vaccines-10-00588-f003:**
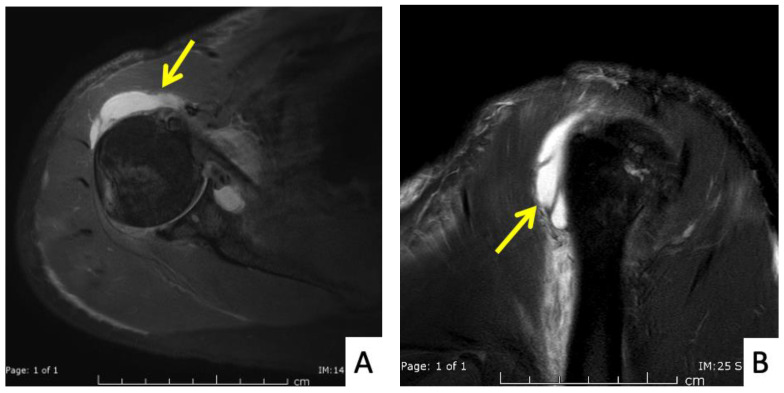
MRI images of the right shoulder of Patient 1; proton density-weighted images in (**A**) axial and (**B**) sagittal views reveal a large amount of fluid in the subdeltoid bursa (yellow arrow). The images also reveal additional subscapularis tendinosis and chondromalacia of the glenohumeral joint, indicating background osteoarthritis.

**Figure 4 vaccines-10-00588-f004:**
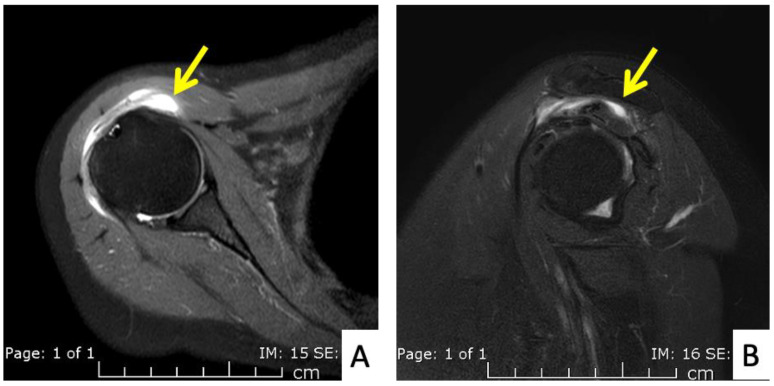
MRI images of the right shoulder of Patient 2; proton density-weighted images in (**A**) axial and (**B**) sagittal views reveal a large amount of fluid in the subacromial-subdeltoid bursa (yellow arrow).

**Figure 5 vaccines-10-00588-f005:**
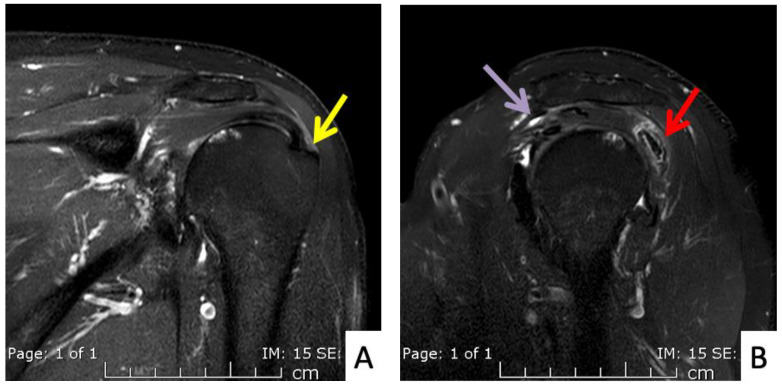
MRI images of the left shoulder of Patient 4; (**A**) coronal and (**B**) sagittal T2-weighted images of the left shoulder reveal a small, partial-thickness, bursal surface tear at the footprint of the supraspinatous tendon (yellow arrow). Increased signal intensity of the myotendinous junction of the infraspinatous tendon indicates muscle strain (red arrow). An edematous subacromial bursa with thin fluid (purple arrow) was also observed.

**Figure 6 vaccines-10-00588-f006:**
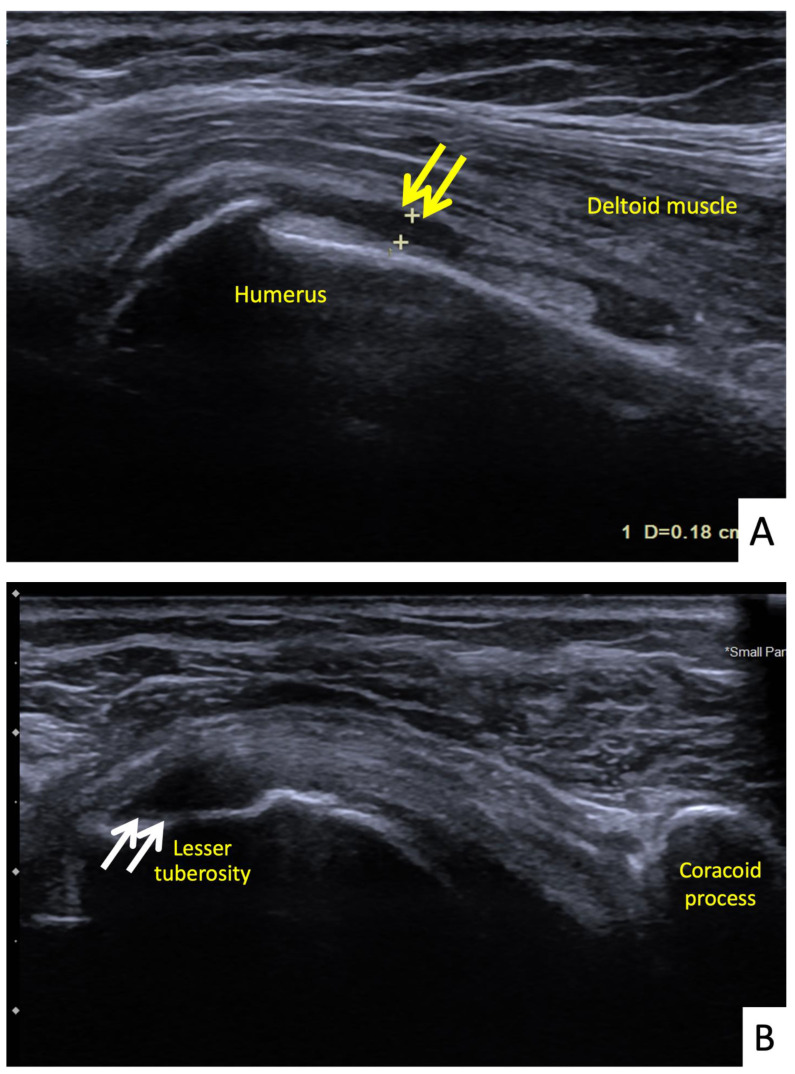
Ultrasonographic images of the right shoulder of Patient 5. (**A**) A longitudinal ultrasonographic image over the lateral aspect of the left proximal humerus with the patient in supination showing a small amount of fluid (yellow arrows) in the mildly distended subdeltoid bursa. (**B**) A transverse ultrasonographic image over the lesser tuberosity of the right shoulder with the patient in the external rotation position showing a partial thickness, articular surface tear at the footprint of the subscapularis tendon (white arrows).

**Figure 7 vaccines-10-00588-f007:**
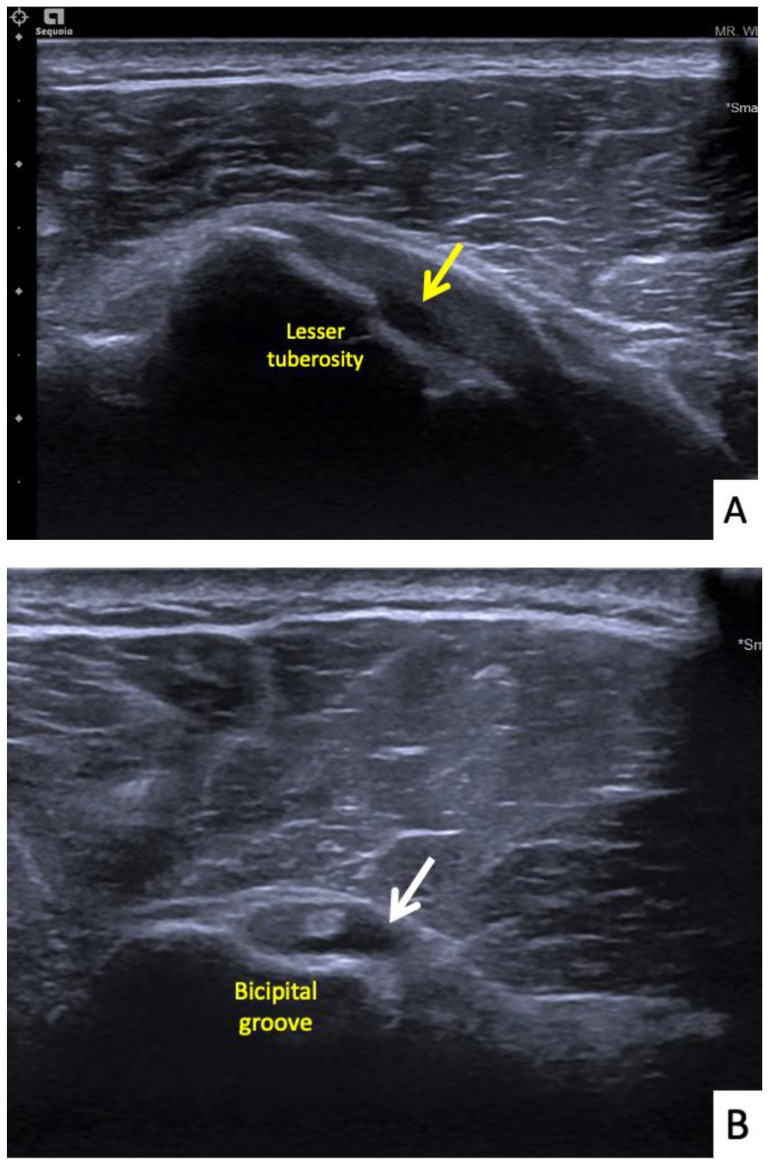
Ultrasonographic images of the right shoulder of Patient 6. (**A**) A transverse ultrasonographic image over the lesser tuberosity of the right shoulder with the patient in the external rotation position showing a small, partial thickness, intrasubstance tear (yellow arrow) at the footprint of the subscapularis tendon. (**B**) A transverse ultrasonographic image over the bicipital groove with the patient placing his hand palm up in supination on his leg showing surrounding fluid (white arrow) within the long head biceps tendon sheath, indicating tenosynovitis.

**Figure 8 vaccines-10-00588-f008:**
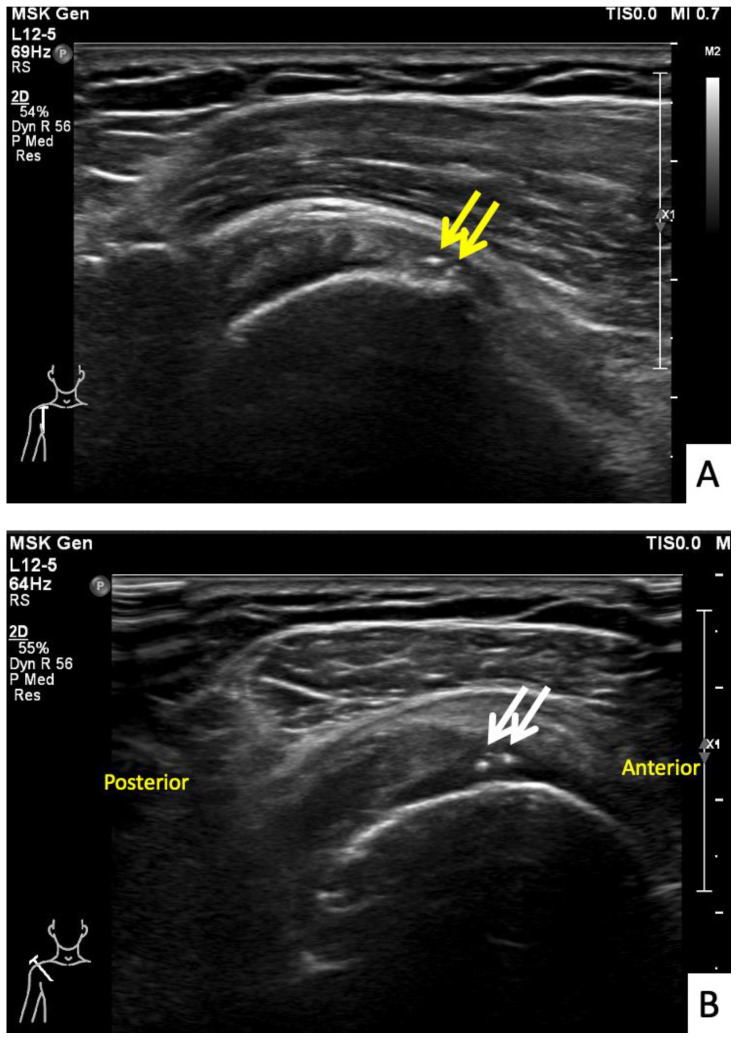
Ultrasonographic images of the right shoulder of Patient 7. (**A**) A sagittal ultrasonographic image over the lesser tuberosity of the right shoulder with the patient in the external rotation position showing a few small calcific deposits (yellow arrows) within the inferior fiber of the subscapularis tendon. (**B**) A transverse ultrasonographic image of the supraspinatous tendon with the modified Crass position of the right shoulder showing a few small calcific deposits (white arrows) within the mid fiber.

**Table 1 vaccines-10-00588-t001:** Characteristics of the study cases (BMI: body mass index, M: male, F: female, TA: triamcinolone acetate).

Case	Age	Sex	BMI (kg/m^2^)	Vaccine	Dose	Onset	Incorrect Technique	Range of Motion	Finding	Medication	Return to Normal Activities after Initial Treatment
1	52	M	24.22	Sinovac	1st	Within 72 h	Wrong direction	Limited	Subacromial bursitis	Antibiotic + oral celecoxib	14 days
2	51	F	23.91	AstraZeneca	1st	3 h	Too high	Limited	Subacromial-subcoracoid bursitis and supraspinatous tear	Oral prednisolone	3 days
3	66	M	24.53	AstraZeneca	2nd	Immediately	Wrong direction	Limited	-	Oral prednisolone	7 days
4	71	M	29.26	AstraZeneca	2nd	24 h	Too high	Full but painful	Thin subacromial-subcoracoid bursitis and low-grade partial tear of supraspinatous tendon	Oral prednisolone	14 days
5	68	F	23.73	AstraZeneca	2nd	Within 24 h	Wrong direction	Full but painful	Thin subdeltoid bursal fluid and partial thickness tear of subscapularis tendon	Oral prednisolone	7 days
6	64	M	26.94	AstraZeneca	1st	Within 48 h	Wrong direction	Limited	Tenosynovitis of long head of biceps and low-grade partial tear of subscapularis tendon	Combined oral prednisolone and TA injection	7 days
7	64	F	23.11	AstraZeneca	1st	Immediately	Too high	Limited	Calcific tendinopathy of supraspinatous tendon	Oral prednisolone	5 days

## Data Availability

All relevant data are included in the manuscript.

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
