# Peer review of "Shoulder Injury Related to COVID-19 Vaccine Administration: A Case Series"

_vaccines, 2022, doi:10.3390/vaccines10040588_

Round 1
Reviewer 1 Report
Well written study
Author Response
Reviewer 1:
Well written study
ANS Thank you for your kind positive comment
Reviewer 2 Report
The manuscript covers an interesting topic regarding possible late complication after covid-19 vaccination, in particular the involvement of shoulder pain.
The topic is relevant and original since COVID-19, and AEFI, are still an important health challenge
It is a case series that improves the statistics regarding this not frequently reported AEFI.
The method is appropriate and clearly reported.
Results are well presented and detailed.
The conclusions are consistent with the evidence and arguments presented.
The references are appropriate.
Tables and figures are appropriate. A graphical abstract might improve the interest for the paper, so please consider adding.
line 272 there is a typo, please correct it.
Author Response
Reviewer 2:
English language and style: Moderate English changes required
ANS This manuscript was proofread by our Faculty English advisor before re-submission.
The method is appropriate and clearly reported.
ANS Thank you.
Results are well presented and detailed.
ANS Thank you.
The conclusions are consistent with the evidence and arguments presented.
ANS Thank you.
The references are appropriate.
ANS Thank you.
Tables and figures are appropriate. A graphical abstract might improve the interest for the paper, so please consider adding.
ANS Thank you for the suggestion. We agree a graphical abstract would be helpful and have added one to the manuscript.
line 272 there is a typo, please correct it.
ANS Thank you for your comment. The typo has been corrected.

Reviewer 3 Report
The paper of Korakot Maliwankul and al. describes a series of 7 cases with a vaccine-related shoulder injury (SIRVAIn all cases of SIRVA, the cause was an incorrect vaccination technique, and patients responded well to oral NSAIDs or oral prednisolone as a treatment. The paper is informative and follows proper form.
However, the manuscript would fit better within the Vaccination Strategies for COVID-19 topic if the publication included information on how the occurrence of a shoulder injury related to vaccine administration and subsequent administration of a SIRVA relates to vaccine efficacy. Do the authors have information on whether patients developed antibodies after this vaccination? And what were the levels in relation to people who were vaccinated correctly?
Could the authors add a paragraph on the efficacy of vaccination in the case of shoulder injury due to incorrect vaccine administration?
And a small remark in the abstract the description of 3 patients was omitted, only male patients were described, similarly in the case series paragraph.
Author Response
Reviewer 3:
However, the manuscript would fit better within the Vaccination Strategies for COVID-19 topic if the publication included information on how the occurrence of a shoulder injury related to vaccine administration and subsequent administration of a SIRVA relates to vaccine efficacy.
ANS No patients caught COVID-19 during the SIRVA treatment and we did not evaluate the antibodies of any of these cases. We did a literature search for antibody studies of SIRVA but we did not find any.
Do the authors have information on whether patients developed antibodies after this vaccination? And what were the levels in relation to people who were vaccinated correctly?
ANS We did not evaluate the level of antibodies in this case series. In the future, we will evaluate the antibodies in cases of SIRVA.
Could the authors add a paragraph on the efficacy of vaccination in the case of shoulder injury due to incorrect vaccine administration? And a small remark in the abstract the description of 3 patients was omitted, only male patients were described, similarly in the case series paragraph.
ANS We are sorry if you misunderstood our basic patient data – in the Abstract we say that 4 of the 7 patients were male, indicating the other 3 were female, and the following average age applied to all 7 patients.